# PilotRAG: Teaching LLMs Multi-Turn Hybrid RAG via Reinforcement Learning

## Abstract

Retrieval-Augmented Generation (RAG) enhances Large Language Models (LLMs) by incorporating external knowledge, typically from unstructured texts or structured graphs. While recent progress has extended text-based RAG to multi-turn reasoning through Reinforcement Learning (RL), existing graph-based and hybrid RAG methods generally rely on fixed or handcrafted multi-turn retrieval procedures rather than an RL-trained policy, and thus do not support adaptive, decision-based multi-turn reasoning. This limitation restricts their ability to incrementally integrate supplementary evidence as reasoning unfolds, thereby reducing their effectiveness on complex multi-hop questions. To address this limitation, we introduce PilotRAG, an RL-based framework that enables LLMs to perform multi-turn and adaptive graph-text hybrid RAG by dynamically interleaving reasoning, hybrid retrieval, and answer formulation. PilotRAG jointly optimizes the entire generation process via RL, allowing the model to learn when to reason, what to retrieve from either unstructured texts or structured graphs, and when to produce final answers, all within a unified generation policy. To guide this learning process, we design a two-stage training framework with a reward function that accounts for both task outcome and retrieval efficiency. By rewarding answer accuracy and efficient retrieval while penalizing redundant retrieval operations, the model learns to retrieve selectively and reason effectively. Experiments on both simple and multi-hop question answering benchmarks demonstrate that PilotRAG significantly outperforms existing RAG baselines, highlighting the benefits of end-to-end RL for enabling adaptive and iterative retrieval in complex reasoning scenarios.

## 1 Introduction

Large Language Models (LLMs) have demonstrated remarkable capabilities in reasoning, decision-making, and long-form generation (Zhao et al., 2023; Touvron et al., 2023; Team et al., 2024), especially when further trained with Reinforcement Learning (RL) (Achiam et al., 2023; Guo et al., 2025; Yang et al., 2025a). These abilities have enabled LLMs to follow complex instructions, emulate chain-of-thought reasoning, and solve complicated multi-hop questions (Zhou et al., 2023; Wei et al., 2022). However, the knowledge of LLMs remains static, bounded by the data available at pretraining time. As a result, LLMs often produce inaccurate or outdated outputs when faced with knowledge-intensive queries that require access to external or up-to-date information (Augenstein et al., 2024; Huang et al., 2025).

To overcome this limitation, Retrieval-Augmented Generation (RAG) has emerged as a core paradigm for enhancing LLMs with access to external knowledge sources (Lewis et al., 2020; Gao et al., 2023). Early RAG systems typically perform a single round of retrieval before generation (Guu et al., 2020; Wang et al., 2023). Recent work has shown the benefits of multi-turn retrieval, where the model interleaves retrieval and reasoning over multiple steps (Yao et al., 2023; Trivedi et al., 2023; Li et al., 2025).

However, these prompt-based approaches often depend on large closed-source models with strong intrinsic reasoning and planning skills. Smaller open-source models struggle to determine when to retrieve, how to formulate retrieval queries, and how to analyze retrieved evidence. This gap has motivated a new line of research (Jin et al., 2025; Song et al., 2025) that employs RL to explicitly

train models to make retrieval and reasoning decisions. By optimizing a learned policy over interleaved thinking and retrieval actions, these RL-based methods aim to equip models with adaptive, context-sensitive retrieval strategies that surpass static instructions.

In parallel, graph-based RAG systems (Edge et al., 2024; Jimenez Gutierrez et al., 2024; Gutiérrez et al., 2025) utilize structured knowledge graphs to integrate and reason over information scattered across multiple passages, thereby improving coverage of factual entities and relations. While graphs enable more accurate entity disambiguation and multi-hop path reasoning than text-only retrieval, retrieving and processing graph evidence is often more computationally expensive, especially in large-scale or dense graphs. Moreover, existing graph-based RAG systems typically operate in a one-shot retrieval setting, fetching graph evidence once before generation, and lack the ability to adaptively choose between graph and text retrieval based on the evolving information needs of the query. Consequently, the current architecture of graph-based RAG systems presents challenges in managing complex reasoning that necessitates multi-turn interactions, and can also lead to unnecessary retrieval overhead when graph access is not essential.

We address these limitations with PILOTRAG, an RL-based framework that enables LLMs to perform multi-turn and hybrid retrieval over both unstructured texts and structured knowledge graphs. Instead of passively executing preset instructions, PILOTRAG takes the role of a pilot that actively orchestrates retrieval decisions, selecting when and where to access external knowledge. To overcome the challenges of managing complex reasoning and avoiding unnecessary retrieval overhead, PILOTRAG learns to interleave reasoning, retrieval, and answer formulation through a unified generation policy, adapting its retrieval behavior to the evolving task context. Instead of relying on static workflow, the model adaptively determines its retrieval behavior based on the evolving context, effectively piloting its interaction with external knowledge in pursuit of accurate and efficient reasoning.

To enable PILOTRAG to generate accurate answers while efficiently retrieving relevant knowledge, we adopt a two-stage Group Relative Policy Optimization (GRPO) (Shao et al., 2024) training framework. In the first stage, the model is rewarded solely for answer correctness, allowing it to acquire the core capability of generating accurate responses and establishing a solid starting point for further optimization. In the second stage, we introduce an additional efficiency reward that discourages unnecessary retrieval, guiding the model to strike a balance between accuracy and computational cost. With these designs, PILOTRAG can achieve both high accuracy and retrieval efficiency in complex multi-hop reasoning tasks.

Our main contributions lie in three aspects:

- We propose PILOTRAG, an RL-based framework for multi-turn and hybrid RAG. The model learns a unified generation policy that interleaves reasoning, adaptive graph-text hybrid retrieval, and answer formulation through a two-stage training framework.

- We design a reward function that jointly optimizes answer accuracy and retrieval efficiency, encouraging the model to retrieve selectively and to reason effectively over retrieved evidence across multiple steps.

- Extensive experiments on five Question Answering (QA) benchmarks demonstrate that PILOTRAG outperforms prior multi-turn and graph-based RAG systems significantly.

## 2 RELATED WORK

### 2.1 RAG

RAG has become a key paradigm for enhancing LLMs with external knowledge, thus mitigating hallucination and improving factual grounding (Guu et al., 2020; Gao et al., 2023). Traditional RAG systems retrieve relevant text chunks from an external knowledge base according to the query, and then feed the query into the LLM together with those text chunks to generate a final answer (Lewis et al., 2020; Yu et al., 2022). Beyond such one-shot retrieve-then-generate pipelines, recent research has explored multi-turn retrieval to provide more fine-grained and incremental supplementation of external knowledge, interleaving reasoning with evidence acquisition. For instance, IRCoT (Trivedi et al., 2023) shows that alternating chain-of-thought reasoning with retrieval improves the performance of LLM on knowledge-intensive multi-hop QA. Search-o1 (Li et al., 2025) further develops

this line by introducing a reason-in-documents module to alleviate the issue of redundant information in retrieved documents.

As an alternative route for deep knowledge integration, graph-based RAG methods incorporate structured knowledge graphs to aggregate evidence across passages and to make relational connections explicit (Peng et al., 2024; Zhao et al., 2023). By exposing entities and relations directly, these methods are particularly effective for multi-hop questions that require linking facts across disparate documents (Edge et al., 2024; Jimenez Gutierrez et al., 2024; Gutiérrez et al., 2025). However, graph retrieval is often more computationally expensive than text retrieval, and existing methods commonly perform one-shot retrieval. Although recent work such as HybGRAG (Lee et al., 2025b) demonstrates that multi-turn hybrid text-graph retrieval is feasible through predefined multi-step procedures, these methods rely on fixed heuristics rather than a learnable policy.

These limitations call for a unified and trainable hybrid framework in which the system can dynamically decide between text and graph retrieval as reasoning unfolds.

## 2.2 RL FOR LLM REASONING

RL has played a central role in improving the reasoning capabilities of LLMs. RL from Human Feedback (RLHF) (Christiano et al., 2017; Ouyang et al., 2022) has established a standard paradigm, where a reward model trained from human preferences directs the optimization of policies (Lambert et al., 2025), allowing models to adhere to instructions and reason with greater accuracy. Proximal Policy Optimization (PPO) (Schulman et al., 2017) remains the predominant algorithm for achieving these goals. More recently, GRPO (Shao et al., 2024) has been proposed as a more efficient variant, which leverages group-wise relative rewards to stabilize training and reduce variance. Building on these advances, researchers have begun to apply RL directly to the training of multi-turn RAG systems (Jin et al., 2025; Song et al., 2025). For instance, Search-R1 (Jin et al., 2025) trains LLMs with RL to decide when and what to search in the middle of reasoning, using only outcome rewards. While this reward design effectively improves correctness, it does not explicitly address retrieval cost or efficiency.

These developments highlight the potential of RL to move beyond static templates, endowing LLMs with adaptive and context-aware retrieval strategies, while also motivating methods that optimize both accuracy and efficiency.

## 3 PILOTRAG

In this section, we present PILOTRAG, an RL-based framework for multi-turn hybrid RAG. We first describe the multi-turn workflow and the mechanism for hybrid knowledge access (Section 3.1). Subsequently, we introduce our two-stage RL framework (Section 3.2), encompassing the formulation of outcome and efficiency rewards, alongside the GRPO-based training algorithm.

### 3.1 OVERALL FRAMEWORK

We begin by outlining the overall architecture of PILOTRAG. This framework integrates LLMs with external retrievers in a multi-turn reasoning loop, where special tokens from the reasoning process can trigger retrieval actions from text and graph knowledge sources. We describe the multi-turn reasoning and retrieval workflow in Section 3.1.1 and the hybrid knowledge access mechanisms of the external retriever in Section 3.1.2.

#### 3.1.1 MULTI-TURN REASONING AND RETRIEVAL WORKFLOW

We formulate multi-turn retrieval-augmented generation as a sequential decision-making process. Given an input query $q$, the policy model $\pi_\theta$ interacts with external knowledge sources over a sequence of steps $b = \{1, \ldots, B\}$, where $B$ is the maximum step budget. At each step, the policy model conditions on the query and the current context to generate an action token. The action space includes continuing internal reasoning, triggering a retrieval operation (`<search> ... </search>`), or producing a final answer (`<answer> ... </answer>`). The retrieval operation further specifies a retrieval mode $m \in \{\texttt{Passage}, \texttt{Graph}, \texttt{Hybrid}\}$ by special tokens

$y \in \{\texttt{[passage]}, \texttt{[graph]}\}$ and a sub-query $q'$, which are used to obtain documents $d$ from the retriever $\mathcal{R}(q', m)$. The retrieved information is then appended to the context and becomes available for subsequent reasoning, as summarized in Algorithm 1.

---

**Algorithm 1** PILOTRAG Framework

---

**Require:** Input query $q$, policy model $\pi_\theta$, retriever $\mathcal{R}$, maximum step budget $B$.
**Ensure:** Final response $y$.
 1: Initialize response $y \leftarrow \emptyset$, step count $b \leftarrow 0$
 2: **while** $b < B$ **do**
 3:    Initialize current rollout $y_b \leftarrow \emptyset$
 4:    **while** True **do**
 5:        Sample next token $y' \sim \pi_\theta(\cdot \mid q, y + y_b)$
 6:        $y_b \leftarrow y_b + y'$
 7:        **if** $y' \in \{\texttt{</search>}, \texttt{</answer>}, \texttt{<eos>}\}$ **then break**
 8:    $y \leftarrow y + y_b$                                   ▷ Combine rollout with history
 9:    **if** $\texttt{<search>} ... \texttt{</search>}$ detected in $y_b$ **then**
10:        **if** $\texttt{[passage]}$ in $y_b$ **then** $m \leftarrow \texttt{Passage}$          ▷ Dense Passage Retrieval
11:        **if** $\texttt{[graph]}$ in $y_b$ **then** $m \leftarrow \texttt{Graph}$
12:        **if** $\texttt{[passage]}$ and $\texttt{[graph]}$ in $y_b$ **then** $m \leftarrow \texttt{Hybrid}$
13:        Extract query $q' \leftarrow \texttt{ParseQuery}(y_b)$
14:        $d \leftarrow \mathcal{R}(q', m)$         ▷ Retrieve documents according to the retrieval mode $m$
15:        Insert into rollout $y \leftarrow y + \texttt{<information>}d\texttt{</information>}$
16:    **else if** $\texttt{<answer>} ... \texttt{</answer>}$ detected in $y_b$ **then**
17:        **return** final response $y$
18:    $b \leftarrow b + 1$
19: **return** final response $y$

---

This workflow enables the model to progressively refine its knowledge state by deciding what to retrieve and when to retrieve it, conditioned on the evolving reasoning trajectory. Moreover, the explicit action space over different retrieval modes allows the model to adaptively balance lightweight passage retrieval with more expensive but structurally intricate graph retrieval, depending on the requirements of the query.

### 3.1.2 HYBRID KNOWLEDGE ACCESS

In PILOTRAG, the retriever $\mathcal{R}$ is responsible for providing external knowledge to support reasoning, with three different retrieval modes.

**Passage Retrieval**. The passage retriever is implemented with Dense Passage Retrieval (DPR) (Karpukhin et al., 2020), which encodes both the sub-query and all passages in the corpus into a shared embedding space. Retrieval is performed by computing similarity scores between the query vector and passage vectors, and the top-$k$ passages are selected as evidence.

**Graph-based Retrieval**. The graph retriever is implemented with HippoRAG 2 (Gutiérrez et al., 2025), which first constructs a knowledge graph over passages. Given a sub-query, the retriever applies personalized PageRank over the graph to propagate relevance from query-linked nodes, thereby identifying passages that are related to the query through multi-hop connections.

**Hybrid Retrieval**. The hybrid retriever combines passage and graph retrieval using Reciprocal Rank Fusion (RRF) (Cormack et al., 2009). Specifically, given two ranked lists, each document is assigned a fused score that decreases with its reciprocal rank in each list, which ensures that documents highly ranked by either retrieval mode are promoted in the merged list. Formally, the fused score is defined as

$$\text{RRF}(d) = \sum_{m \in \{\texttt{passage}, \texttt{graph}\}} \frac{1}{k + \text{rank}_m(d)}, \tag{1}$$

where $\text{rank}_m(d)$ denotes the rank position of document $d$ in retrieval mode $m$, and $k$ is a smoothing hyperparameter. Documents are then re-ranked according to $\text{RRF}(d)$ to form the final hybrid list.

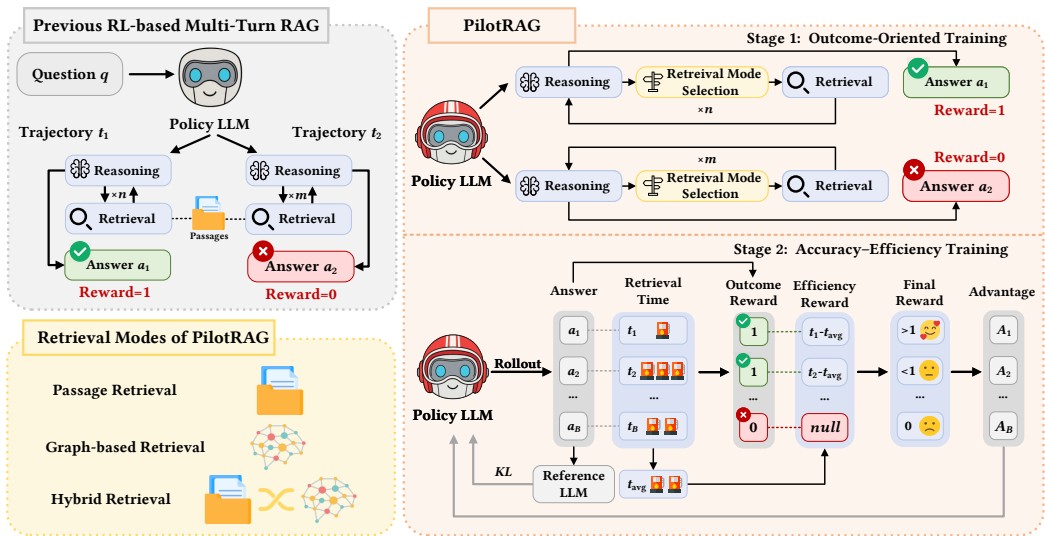

Figure 1: Previous RL-based multi-turn RAG vs. PILOTRAG. Prior methods focus on interleaving reasoning with passage retrieval and reward on answer correctness. PILOTRAG extends retrieval to passage, graph, and hybrid modes, and is trained with a two-stage RL framework that optimizes both accuracy and efficiency.

## 3.2 TWO-STAGE REINFORCEMENT LEARNING

To optimize the unified generation policy, PILOTRAG is trained with a two-stage RL framework based on GRPO. The motivation is to first ensure that the model acquires the basic ability to produce correct answers, and then to further refine its retrieval strategy to improve efficiency without sacrificing accuracy, as shown in Figure 1. In this section, we introduce the reward design that guides the learning objectives (Section 3.2.1) and the training algorithm that realizes the optimization procedure (Section 3.2.2).

### 3.2.1 REWARD DESIGN

RL optimization is fundamentally guided by the reward signal. To support the two-stage training, we devise different rewards for each stage, i.e., outcome-oriented reward and accuracy–efficiency reward.

**Stage 1: Outcome-Oriented Reward.** In the first stage, the reward is defined purely by the correctness of the model output. Specifically, the reward is set to 1 if the generated answer $y$ exactly matches the ground-truth label $y^*$, and 0 otherwise:

$$R_\phi(x, y) = \text{EM}(y, y^*). \tag{2}$$

**Stage 2: Accuracy–Efficiency Reward.** In the second stage, we extend the reward function to jointly optimize for correctness and retrieval efficiency. The reward is defined as

$$R_\phi(x, y) = \begin{cases} R_{\text{outcome}}, & \text{if } R_{\text{outcome}} = 0 \\ R_{\text{outcome}} + R_{\text{efficiency}}, & \text{if } R_{\text{outcome}} = 1 \end{cases}, \tag{3}$$

where $R_{\text{outcome}} \in \{0, 1\}$ denotes exact match accuracy. The efficiency reward $R_{\text{efficiency}}$ is computed from the total retrieval time across all reasoning steps. We apply a centered scaling by subtracting the average retrieval time $t_{\text{avg}}$, such that

$$R_{\text{efficiency}} = \frac{t_{\text{avg}} - t}{T}, \tag{4}$$

where $t$ is the total retrieval time for the current trajectory, $t_{\text{avg}}$ is the average retrieval time of the current batch, and $T$ is a normalization constant ensuring the value lies in $[0, 0.5]$. This design

provides positive reward for trajectories that achieve the correct answer at a pace exceeding the average, while imposing penalties on those that do not, thereby encouraging the model to retrieve more selectively without sacrificing answer quality.

### 3.2.2 TRAINING ALGORITHM

We adopt GRPO (Shao et al., 2024; Guo et al., 2025) to train the unified generation policy $\pi_\theta$ over interleaved reasoning and retrieval actions. GRPO stabilizes learning by comparing trajectories within a group, thereby reducing variance in sparse-reward settings.

The policy model $\pi_\theta$ is optimized by maximizing the following objective:

$$
\begin{aligned}
J_{\text{GRPO}}(\theta) = & \mathbb{E}_{x \sim Q, \{y_i\}_{i=1}^G \sim \pi_{\theta_{\text{old}}}(Y|q)} \\
& \frac{1}{G} \sum_{i=1}^G \left[ \min\left( r_i(\theta) A_i, \text{clip}(r_i(\theta), 1 - \epsilon, 1 + \epsilon) A_i \right) - \beta \, \mathbb{D}_{\text{KL}}[\pi_{\theta_{\text{old}}} \| \pi_\theta] \right],
\end{aligned}
\tag{5}
$$

where $\epsilon$ and $\beta$ are hyperparameters, $\pi_{\theta_{\text{old}}}$ denotes the old policy, $r_i(\theta) = \frac{\pi_\theta(y_i|x)}{\pi_{\theta_{\text{old}}}(y_i|x)}$, $A_i$ denotes the *group-relative advantage* for the $i$-th trajectory, and the KL penalty $\mathbb{D}_{\text{KL}}[\pi_{\theta_{\text{old}}} \| \pi_\theta]$ regularizes the new policy against deviating excessively from the old policy. Further theoretical analysis on the effectiveness of the efficiency reward and GRPO is provided in Appendix A.

## 4 EXPERIMENTS

### 4.1 EXPERIMENTAL SETTING

**Evaluation Datasets.** Following Jimenez Gutierrez et al. (2024) and Gutiérrez et al. (2025), we evaluate PILOTRAG on five widely used benchmarks for simple and multi-hop QA, namely PopQA (Mallen et al., 2023), Natural Questions (NQ) (Kwiatkowski et al., 2019; Wang et al., 2024), HotpotQA (Yang et al., 2018), 2WikiMultihopQA (2Wiki) (Ho et al., 2020), and MuSiQue (Trivedi et al., 2022). PopQA is an open-domain QA dataset designed to evaluate factual recall over long-tail knowledge, and NQ contains naturally occurring queries paired with answers from Wikipedia. HotpotQA and 2WikiMultihopQA focus on multi-hop reasoning across Wikipedia passages, while MuSiQue requires reasoning over compositional sub-questions. The statistics of these datasets are shown in Table 1.

Table 1: Dataset statistics

|                    | PopQA | NQ    | HotpotQA | 2Wiki | MuSiQue |
|--------------------|-------|-------|----------|-------|---------|
| Number of queries  | 1,000 | 1,000 | 1,000    | 1,000 | 1,000   |
| Number of passages | 8,678 | 9,633 | 9,811    | 6,119 | 11,656  |

**Baselines.** We compare PILOTRAG against several types of representative approaches: (1) **Vanilla RAG** (Lewis et al., 2020), which performs single-shot dense passage retrieval and generation. (2) Multi-turn RAG methods, including **Search-o1** (Li et al., 2025) and **Search-R1** (Jin et al., 2025), wherein the latter utilizes RL to enhance multi-turn passage RAG. (3) graph-based RAG methods, including **GraphRAG** (Edge et al., 2024), **LightRAG** (Guo et al., 2024), **RAPTOR** (Sarthi et al., 2024), **HippoRAG** (Jimenez Gutierrez et al., 2024), and **HippoRAG 2** (Gutiérrez et al., 2025), which leverage structured knowledge graphs for retrieval.

**Implementation Details.** We conduct training using Qwen2.5-3B-Instruct (Yang et al., 2025b) as the backbone model. The training data consists of 10k sampled queries from the HotpotQA training set (Yang et al., 2018), while the retrieval corpus is built from their associated documents. For retrieval, we adopt Contriever (Izacard et al., 2022) as the dense retriever. To ensure comparability across methods, the retrieval budget $B$ is fixed at 4 turns, and the number of retrieved passages per call is set to $k = 3$. PILOTRAG is optimized using the two-stage RL procedure described in Section 3.2. Stage 1 is trained for 0.5 epoch with EM-based rewards only. Stage 2 continues for an additional 0.5 epoch. For baseline evaluations, text-based RAG systems are assessed under

| Method | Simple QA | | | | Multi-hop QA | | | | | | Average | |
| | PopQA | | NQ | | HotpotQA | | 2Wiki | | MuSiQue | | | |
| | EM | F1 | EM | F1 | EM | F1 | EM | F1 | EM | F1 | EM | F1 |
|---|---|---|---|---|---|---|---|---|---|---|---|---|
| *GPT-4o-mini* | | | | | | | | | | | | |
| Direct Inference | 16.1 | 22.7 | 35.2 | 52.7 | 28.6 | 41.0 | 30.2 | 36.3 | 11.2 | 22.0 | 24.3 | 34.9 |
| *Graph-based RAG* | | | | | | | | | | | | |
| GraphRAG | 30.7 | 51.3 | _38.0_ | _55.5_ | _51.4_ | _67.6_ | 45.7 | 61.0 | 27.0 | _42.0_ | 38.6 | _55.5_ |
| LightRAG | 1.9 | 14.8 | 2.8 | 15.4 | 9.9 | 20.2 | 2.5 | 12.1 | 2.0 | 9.3 | 3.8 | 14.4 |
| RAPTOR | _41.9_ | 55.1 | 37.8 | 54.5 | 50.6 | 64.7 | 39.7 | 48.4 | _27.7_ | 39.2 | 39.5 | 52.4 |
| HippoRAG | **42.5** | **56.2** | 37.2 | 52.5 | 46.3 | 60.0 | _59.4_ | _67.3_ | 24.0 | 35.9 | _41.9_ | 54.4 |
| HippoRAG v2 | 41.7 | _55.7_ | **43.4** | **60.0** | **56.3** | **71.1** | **60.5** | **69.7** | **35.0** | **49.3** | **47.4** | **61.2** |
| *Qwen2.5-3B-Instruct* | | | | | | | | | | | | |
| Vanilla RAG | 30.3 | 41.6 | 18.1 | 31.8 | 29.5 | 41.8 | 19.7 | 27.4 | 10.3 | 17.5 | 21.6 | 32.0 |
| *Graph-based RAG* | | | | | | | | | | | | |
| HippoRAG v2 | 29.1 | 40.1 | 20.3 | 33.5 | 31.2 | 45.0 | 21.5 | 33.8 | 12.2 | 20.2 | 22.9 | 34.5 |
| *Multi-turn RAG* | | | | | | | | | | | | |
| Search-o1 | 17.1 | 23.8 | 19.9 | 29.1 | 18.7 | 26.3 | 16.9 | 20.9 | 3.9 | 10.5 | 15.3 | 22.1 |
| Search-R1 | _45.8_ | 53.3 | **46.2***  | **54.8*** | _45.2_* | _56.9_* | 42.4 | 50.8 | _22.2_ | 30.9 | 40.4 | 49.3 |
| **PILOTRAG (ours)** | **49.4** | **56.8** | _44.1_ | _53.4_ | **53.2*** | **65.1*** | **57.5** | **64.1** | **30.7** | **39.3** | **47.0** | **55.7** |

Table 2: Main results on simple and multi-hop QA benchmarks. The best results within each backbone group are indicated in bold, while the underlined values represent the second-best results. * represents in-domain datasets.

the same Qwen2.5-3B-Instruct backbone, while graph-based RAG systems utilize the GPT-4o-mini backbone. In particular, HippoRAG v2 (Gutiérrez et al., 2025), the strongest graph-based baseline, is evaluated employing both Qwen2.5-3B-Instruct and GPT-4o-mini backbones. For the strongest text-based baseline, Search-R1 (Jin et al., 2025), we directly use their released GRPO-trained model on Qwen2.5-3B-Instruct, which is trained on 170k samples from NQ and HotpotQA, thereby possessing a considerably larger training set in comparison to our 10k-sample dataset. We report Exact Match (EM) and F1 scores as evaluation metrics. Additional implementation details, including the training prompt template, hyperparameters, and training configuration, are provided in Appendix B.

## 4.2 MAIN RESULT

We conduct a comprehensive comparison of PILOTRAG against all the baseline methods, as shown in Table 6. From the results, we make the following key observations:

**(1) PILOTRAG substantially improves the performance of a small backbone, especially on multi-hop QA.** Graph-based methods such as HippoRAG v2 perform well with the strong GPT-4o-mini backbone but drop sharply with the smaller Qwen2.5-3B-Instruct, indicating that small LLMs struggle to handle complex reasoning chains. In contrast, PILOTRAG achieves much better performance on this small backbone by jointly learning reasoning, retrieval, and answer generation within a unified policy model. This shows that explicitly training the full decision process enables small LLMs to effectively execute multi-turn reasoning and retrieval, which traditional graph-based methods fail to elicit.

**(2) PILOTRAG approaches GPT-4o-mini-based graph-based RAG systems despite using a much smaller model.** Despite the large performance gap usually observed between GPT-4o-mini and Qwen2.5-3B-Instruct, PILOTRAG narrows this gap substantially and even surpasses several graph-based systems built on GPT-4o-mini. This suggests that improving the policy for coordinating reasoning and retrieval can be as impactful as scaling up the backbone itself. In particular, by training a unified policy model rather than only improving retrieval quality or graph coverage, PILOTRAG allows a small open-source model to approximate the reasoning behavior of much stronger proprietary LLMs.

| Method | Simple QA | | | | Multi-hop QA | | | | | | Average | |
| | PopQA | | NQ | | HotpotQA | | 2Wiki | | MuSiQue | | | |
| | EM | F1 | EM | F1 | EM | F1 | EM | F1 | EM | F1 | EM | F1 |
|---|---|---|---|---|---|---|---|---|---|---|---|---|
| **PILOTRAG** | **49.4** | **56.8** | **44.1** | **53.4** | 53.2 | 65.1 | **57.5** | **64.1** | **30.7** | **39.3** | **47.0** | **55.7** |
| w/o efficiency reward | 41.5 | 54.1 | 41.1 | 51.7 | **53.7** | **66.2** | 56.9 | 65.0 | 30.2 | 38.9 | 44.7 | 55.2 |
| w/o training | 26.2 | 41.3 | 18.6 | 30.5 | 35.4 | 46.9 | 24.4 | 37.9 | 15.4 | 24.4 | 24.0 | 36.2 |

Table 3: Ablation on RL training. "w/o training" denotes the base model without any RL training, and "w/o efficiency reward" denotes training only with EM-based outcome rewards.

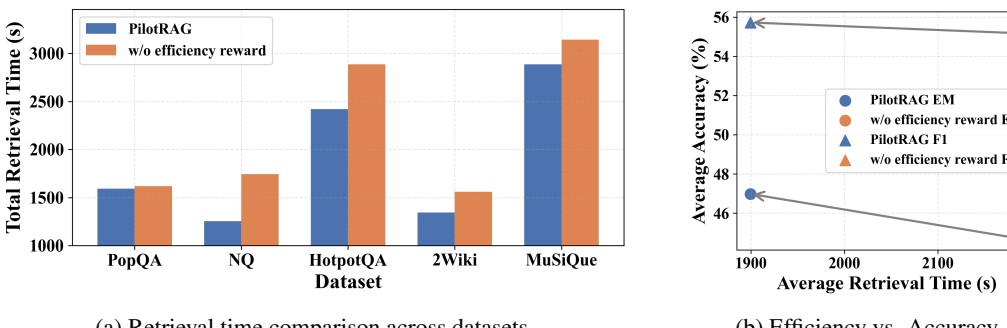

(a) Retrieval time comparison across datasets.

(b) Efficiency vs. Accuracy.

Figure 2: Performance comparison of PILOTRAG and its variant without efficiency reward. (a) Comparing total retrieval time across five datasets. (b) The average retrieval time vs. average EM and F1.

**(3) PILOTRAG outperforms the strongest RL-trained multi-turn baseline with much smaller training cost.** Search-R1, the strongest prior RL-based multi-turn system, jointly trains reasoning, retrieval query generation, and answer generation on 170k questions from NQ and HotpotQA. In contrast, PILOTRAG is trained to additionally exploit graph-based retrieval and to dynamically select among passage, graph, and hybrid retrieval modes based on the task context. These capabilities enable it to construct more comprehensive evidence and adapt its retrieval strategy to different question types. Despite being trained on a mere 10k HotpotQA instances, PILOTRAG outperforms Search-R1 across all datasets, with the exception of NQ, their in-domain dataset, demonstrating that structured retrieval and retrieval mode selection can yield more effective and sample-efficient multi-turn RAG policies than scaling training data alone.

## 4.3 DETAILED ANALYSIS

In this section, we analyze the learning dynamics of PILOTRAG, highlighting how it simultaneously preserves effectiveness, improves retrieval efficiency, and handles more complex reasoning chains.

### 4.3.1 PILOTRAG LEARNS TO AVOID SACRIFICING EFFECTIVENESS

We first examine whether incorporating efficiency-aware rewards compromises the effectiveness of PILOTRAG. Table 3 compares the full model against two ablated variants: (1) a model trained only with outcome rewards from Stage 1, and (2) the untrained backbone. To ensure a fair comparison, the Stage 1-only variant is trained for one full epoch with outcome rewards, while PILOTRAG adopts a two-stage schedule with 0.5 epoch on outcome rewards followed by 0.5 epoch on accuracy–efficiency rewards, resulting in the same total number of training steps.

The results show that PILOTRAG maintains comparable or even higher accuracy than its Stage 1 counterpart, while both substantially outperform the untrained model. This indicates that introducing the efficiency-aware objective does not trade off answer correctness for reduced retrieval cost. Instead, the model continues to improve or preserve task performance as it learns to optimize re-

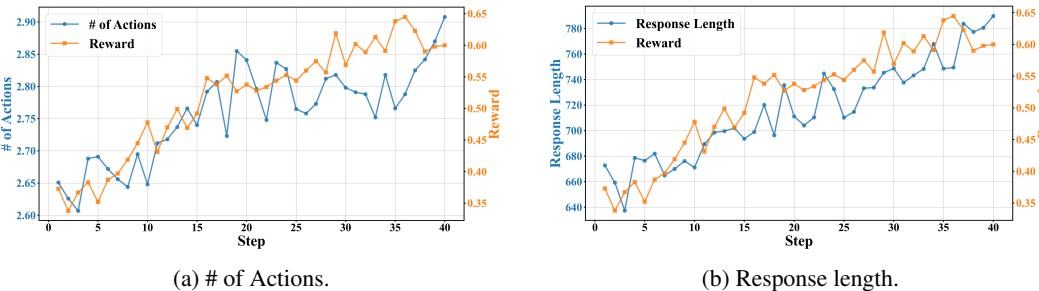

(a) # of Actions.

(b) Response length.

Figure 3: Training dynamics of PILOTRAG, showing the evolution of key performance indicators throughout training. (a) Number of actions vs. Reward. (b) Response length vs. Reward.

trieval behaviors, demonstrating that efficiency can be enhanced without sacrificing effectiveness. A further analysis of the retrieval modes and effectiveness can be found at Appendix C.

### 4.3.2 PILOTRAG LEARNS TO RETRIEVE EFFICIENTLY

We evaluate whether PILOTRAG learns to retrieve evidence more efficiently by comparing it against a variant trained with outcome reward only. Figure 2a shows that PILOTRAG consistently reduces total retrieval time across all datasets, with the largest savings observed on NQ and HotpotQA. This demonstrates that incorporating efficiency rewards encourages the policy to avoid unnecessary retrieval steps while still collecting sufficient evidence. Figure 2b further shows the trade-off between retrieval time and task accuracy. Arrows indicate the movement from the variant without efficiency reward to PILOTRAG, highlighting that efficiency gains do not come at the cost of answer quality. Mathematically, this phenomenon is captured by the normalized advantage function $A_i$ (Equation (9)), which balances outcome and efficiency rewards. A comprehensive theoretical analysis is available in Appendix A.4.

These results confirm that PILOTRAG can effectively optimize retrieval efficiency while maintaining high EM and F1 scores, validating the effectiveness of the two-stage RL training with batch-level efficiency rewards and GRPO.

### 4.3.3 PILOTRAG LEARNS TO HANDLE COMPLEX REASONING CHAINS

To better understand how PILOTRAG evolves during training, we track the dynamics of several behavioral and performance indicators, as shown in Figure 3. At the early stages of training, the model exhibits limited reasoning ability, often producing short responses with only a few actions and achieving low rewards. As training progresses, both the average number of actions and the average response length gradually increase alongside the reward signal. This trend suggests that the model is learning to construct more elaborate reasoning trajectories and to leverage multi-turn interactions more effectively, which in turn leads to higher task performance. For a qualitative view of how this behavioral shift manifests in practice, we present a set of case studies comparing model outputs before and after training in Appendix E.

## 5 CONCLUSIONS

In this paper, we presented PILOTRAG, an RL framework for multi-turn hybrid RAG. Unlike prior multi-turn RAG systems that rely on static prompting or single-mode retrieval, our approach learns a unified policy that interleaves reasoning, retrieval mode selection, retrieval query generation, and answer generation. By explicitly modeling retrieval actions and dynamically selecting between passage, graph-based, and hybrid retrieval, PILOTRAG enables fine-grained control over knowledge access in complex reasoning tasks. Our two-stage training framework further ensures that the model first acquires robust answer correctness and then improves retrieval efficiency without sacrificing accuracy. Experiments conducted on five knowledge-intensive QA benchmarks demonstrate that PILOTRAG significantly outperforms existing graph-based and multi-turn RAG systems, highlighting that efficiency gains can be achieved without compromising answer quality.

ETHICS STATEMENT

This work aims to advance the development of multi-turn hybrid RAG through RL. Our primary objective is to enhance factual accuracy and retrieval efficiency in complex reasoning tasks, with datasets utilized for both training and evaluation purposes that are publicly available.

REPRODUCIBILITY STATEMENT

In the appendix, we provide detailed descriptions of the model architecture, training hyperparameters, dataset preprocessing, and evaluation protocols. To facilitate replication, we release all code, configuration files, and training scripts in an anonymized GitHub repository: `https://anonymous.4open.science/r/PilotRAG`. All experiments were conducted with publicly available datasets. In addition, we will release the trained models upon acceptance of this paper, enabling researchers to reproduce our results and build upon our framework.

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

## A  THEORETICAL ANALYSIS OF EFFICIENCY REWARD

In this section, we provide a detailed theoretical analysis of why the efficiency reward designed in PILOTRAG improves selective retrieval in GRPO-based training.

### A.1  BATCH-LEVEL EFFICIENCY REWARD

Let $\tau_i$ denote the $i$-th trajectory in a group of $G$ trajectories sampled from a batch of size $B$. The total reward for trajectory $\tau_i$ is

$$R_\phi(\tau_i) = R_{\text{outcome}}(\tau_i) + R_{\text{efficiency}}(\tau_i), \qquad (6)$$

where $R_{\text{outcome}}(\tau_i) \in \{0, 1\}$ indicates the correctness of the answer.

The efficiency reward is centered on the batch-level average retrieval time rather than the group-level average:

$$R_{\text{efficiency}}(\tau_i) = \frac{t_{\text{avg}} - t_i}{T}, \quad t_{\text{avg}} = \frac{1}{B} \sum_{\tau \in \text{batch}} t_\tau, \qquad (7)$$

where $t_i$ is the total retrieval time of trajectory $\tau_i$, and $T$ is a normalization constant.

There are three main reasons for choosing batch-level efficiency reward together with GRPO-based training, instead of using group-level efficiency:

- **Variance reduction and stability.** Batch-level averaging reduces the impact of noisy fluctuations in retrieval time (e.g., hardware latency or network delays).
- **Mitigating anomalies across queries.** Although batch-level normalization may produce unusually high or low raw efficiency rewards for certain queries, GRPO's group-relative advantage compensates for this effect.
- **Encouraging selective retrieval.** Combining batch-level centering with GRPO advantage ensures that trajectories with unnecessary retrieval are penalized, while efficient yet accurate trajectories are favored.

In the following, we analyze each of these points in detail.

### A.2  VARIANCE REDUCTION AND STABILITY

Each GRPO group may contain only a few trajectories (e.g., $G = 5$). Raw retrieval times $t_i$ can fluctuate due to hardware noise, network latency, or retriever stochasticity. If group-level averaging were used, such fluctuations could lead to unstable rewards. By computing $t_{\text{avg}}$ across the entire batch, we obtain a more stable reference signal that smooths out these random variations.

This reduces the variance of the group-relative advantage, which in turn stabilizes policy gradient updates.

### A.3  MITIGATING ANOMALIES ACROSS QUERIES

Batch-level normalization may produce unusually high or low raw efficiency rewards for certain queries. For example, a simple question requiring little retrieval could yield a disproportionately large positive $R_{\text{efficiency}}(\tau_i)$. This is indeed a potential drawback of using batch-level efficiency reward.

However, GRPO compensates for this issue through its group-relative formulation. Although $R_{\text{efficiency}}(\tau_i)$ is normalized at the batch level, the advantage $A_i$ is computed relative to the group mean reward within each GRPO group, i.e.,

$$A_i \;=\; \frac{R_\phi(\tau_i) - \frac{1}{G} \sum_{j=1}^{G} R_\phi(\tau_j)}{\text{std}\big(\{R_\phi(\tau_j)\}_{j=1}^{G}\big)}. \qquad (8)$$

Even if a particular query obtains an abnormally large batch-normalized efficiency reward, its influence on learning is moderated by this group-relative centering. As a result, the combination of batch-level normalization and group-level centering ensures that the learning signal remains consistent across diverse query types.

## A.4 ENCOURAGING SELECTIVE RETRIEVAL

For trajectories that correctly answer the query ($R_{\text{outcome}} = 1$), the numerator of $A_i$ can be decomposed into outcome and efficiency components, yielding

$$A_i = \frac{\left(1 - \overline{R_{\text{outcome}}}\right) + \left(R_{\text{efficiency}}(\tau_i) - \overline{R_{\text{efficiency}}}\right)}{\text{std}\left(\{R_\phi(\tau_j)\}_{j=1}^G\right)}, \tag{9}$$

where $\overline{R_{\text{outcome}}}$ and $\overline{R_{\text{efficiency}}}$ are group means computed within the GRPO group, while $R_{\text{efficiency}}(\tau_i)$ itself is computed using the batch-level reference $t_{\text{avg}}$.

From Equation (9) we see:

- If a trajectory answers correctly and its retrieval time is less than the group average, then $R_{\text{efficiency}}(\tau_i) - \overline{R_{\text{efficiency}}} > 0$, and consequently, the numerator increases, thereby rewarding the policy for selective retrieval.

- If a trajectory performs redundant retrieval, the efficiency term is negative and reduces $A_i$, discouraging unnecessary retrieval.

Thus, GRPO guides the policy towards trajectories that balance correctness with efficient retrieval.

# B IMPLEMENTATION DETAILS

## B.1 TRAINING PROMPT TEMPLATE

The training prompt template for the policy LLM is shown in Table 4.

Table 4: Training prompt template for multi-turn reasoning and retrieval.

---

**Training Prompt Template for the Policy LLM**

Answer the given question. You must conduct reasoning inside `<think>` and `</think>` first every time you get new information. After reasoning, if you find you lack some knowledge, you can call a search engine using the following strict format:
1. You MUST first decide which retrieval mode to use (both modes will return relevant documents, but use different retrieval methods):

- Use `[passage]` to find documents using semantic similarity-based dense retrieval

- Use `[graph]` to find documents through graph-based retrieval, which performs retrieval on a structured knowledge graph constructed from documents using fact ranking and graph reasoning

- You can combine them as `[graph][passage]` to get documents from both retrieval methods

2. Then formulate your specific search query based on what information you need
3. Finally, wrap everything in `<search>` and `</search>` tags
For example:

- Using dense retrieval: `<search> [passage]` the capital of France `</search>`

- Using graph-based retrieval: `<search> [graph]` the capital of France `</search>`

- Using both methods: `<search> [graph][passage]` the capital of France `</search>`

The search results (relevant documents) will be returned between `<information>` and `</information>` tags. You can search as many times as you want. If you find no further external knowledge needed, you can directly provide the answer inside `<answer>` and `</answer>`, without detailed illustrations. For example, `<answer>` Paris `</answer>`.
Question: {question}

---

### B.2 TRAINING DETAILS

**Hyperparameters.** For GRPO training of PILOTRAG, we set the policy LLM learning rate to $1 \times 10^{-6}$, a total batch size of 256, with a mini-batch size of 128 and a micro-batch size of 32. The KL divergence regularization coefficient $\beta$ is set to 0.001, and the clip ratio $\epsilon$ is set to 0.2. The retrieval budget is fixed at $B = 4$, and the number of retrieved passages per call is $k = 3$. The maximum sequence length is set to 4,096 tokens, with a maximum response length of 500 tokens, a maximum start length of 2,048 tokens, and a maximum observation length of 500 tokens.

**Training Configuration.** Our training framework is adapted from the Search-R1 training framework (Jin et al., 2025), which builds upon the verl (Sheng et al., 2025). Training is conducted on a single node with 8×80GB NVIDIA A100 GPUs. Two GPUs are allocated for the retrieval service, and four GPUs are used for model training. To improve memory efficiency, we enable gradient checkpointing and apply Fully Sharded Data Parallel (FSDP) with CPU offloading for parameters, gradients, and optimizer states. Rollouts are sampled with vLLM (Kwon et al., 2023) using a tensor parallel size of 1 and a GPU memory utilization ratio of 0.6. The rollout sampling temperature is set to 1.0.

**Two-Stage Training.** Stage 1 is trained for 20 steps (0.5 epoch) with EM-based rewards only, ensuring correctness. Stage 2 continues for an additional 20 steps (0.5 epoch) with the efficiency-aware reward introduced in Section 3.2. In both stages, we sample five responses per prompt during training to compute group-relative advantages. Checkpoints are saved every 10 steps, and the final checkpoint is used for evaluation.

**Training Process.** Figure 4 illustrates the evolution of key performance indicators during training, including EM score, batch-level average retrieval time, and validation score, along with reward signals. As training progresses, all three indicators exhibit a consistent upward trend, along with a steady increase in rewards. These results indicate that the two-stage RL framework effectively guides the model toward more accurate behaviors, while maintaining robust generalization to unseen validation data.

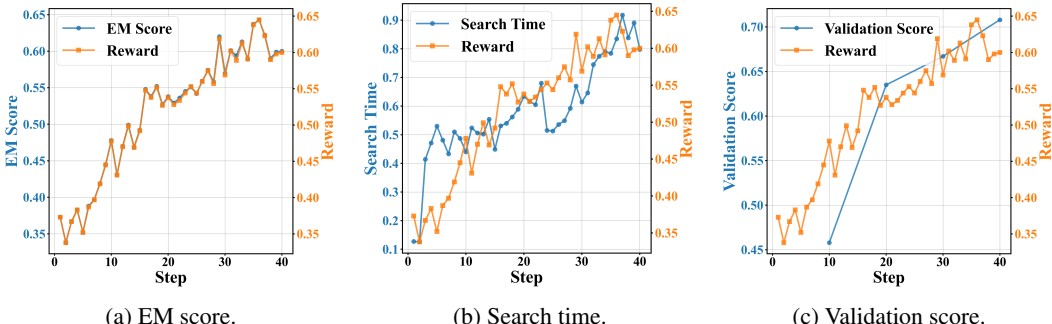

|  |  |  |
|---|---|---|
| (a) EM score. | (b) Search time. | (c) Validation score. |

Figure 4: Training process of PILOTRAG, showing the evolution of key performance indicators throughout training. (a) EM score vs. Reward. (b) Search time vs. Reward. (c) Validation score vs. Reward.

## C ANALYSIS OF RETRIEVAL MODES

Table 5 presents an ablation study comparing PILOTRAG with variants restricted to a single retrieval mode. From the results, we make several observations:

**(1) Complementary strengths of text and graph retrieval.** Passage-only retrieval performs competitively on simple QA benchmarks such as PopQA and NQ, where single-document evidence is often sufficient. In contrast, graph-only retrieval is particularly advantageous on multi-hop QA datasets such as HotpotQA and 2Wiki, where relational structures facilitate entity disambiguation and multi-step reasoning.

| Method | Simple QA | | | | Multi-hop QA | | | | | | Average | |
|---|---|---|---|---|---|---|---|---|---|---|---|---|
| | PopQA | | NQ | | HotpotQA | | 2Wiki | | MuSiQue | | | |
| | EM | F1 | EM | F1 | EM | F1 | EM | F1 | EM | F1 | EM | F1 |
| PILOTRAG | **49.4** | **56.8** | **44.1** | **53.4** | 53.2 | 65.1 | **57.5** | **64.1** | **30.7** | **39.3** | **47.0** | **55.7** |
| w/ only passage retrieval | 48.7 | 55.2 | 43.9 | 53.3 | 53.1 | 64.5 | 53.0 | 58.9 | 28.0 | 36.2 | 45.3 | 53.6 |
| w/ only graph-based retrieval | 49.3 | **56.8** | 43.8 | 53.0 | **53.4** | 65.5 | 57.4 | 64.2 | 29.7 | 38.3 | 46.7 | 55.6 |
| w/ only hybrid retrieval | 48.7 | 55.9 | 43.1 | 52.5 | 53.3 | 65.1 | 53.7 | 59.8 | 28.8 | 37.0 | 45.5 | 54.0 |

Table 5: Ablation on retrieval mode. We compare PILOTRAG with variants restricted to only passage retrieval, only graph retrieval, or only hybrid retrieval.

| Method | Simple QA | | | | Multi-hop QA | | | | | | Average | |
|---|---|---|---|---|---|---|---|---|---|---|---|---|
| | PopQA | | NQ | | HotpotQA | | 2Wiki | | MuSiQue | | | |
| | EM | F1 | EM | F1 | EM | F1 | EM | F1 | EM | F1 | EM | F1 |
| *Qwen2.5-7B* | | | | | | | | | | | | |
| *Graph-based RAG* | | | | | | | | | | | | |
| HippoRAG v2 | 27.0 | 37.9 | 8.1 | 27.2 | 27.4 | 46.1 | 16.8 | 34.7 | 12.3 | 24.0 | 18.3 | 34.0 |
| *Multi-turn RAG* | | | | | | | | | | | | |
| Search-o1 | 4.7 | 7.2 | 18.1 | 27.5 | 13.5 | 19.1 | 6.4 | 7.9 | 2.9 | 7.7 | 9.1 | 13.9 |
| R1-Searcher | 28.4 | 41.0 | 41.6 | 52.2 | 46.6* | 56.7* | 41.7* | 49.0* | 29.3 | 37.6 | 37.5 | 47.3 |
| Search-R1 | **51.3** | **57.1** | **56.8*** | **65.3*** | 51.0* | 62.0* | 51.8 | 58.9 | 32.0 | 40.8 | 48.6 | 56.8 |
| PILOTRAG (ours) | 50.6 | 56.4 | 51.5 | 60.4 | **60.8*** | **72.5*** | **57.1** | **64.6** | **39.6** | **49.3** | **51.9** | **60.6** |

Table 6: Results on methods with Qwen2.5-7B backbone. The best results are indicated in bold, while the underlined values represent the second-best results. * represents in-domain datasets.

**(2) Limitations of indiscriminate hybrid retrieval.** Forcing the model to always use hybrid retrieval (i.e., consulting both passages and graphs simultaneously) yields lower performance than graph-only retrieval. This suggests that indiscriminate combination of heterogeneous sources introduces noise and inefficiency, and that adaptivity in retrieval choice is essential.

**(3) Advantages of adaptive retrieval in PILOTRAG.** The full PILOTRAG, which dynamically selects between text, graph, and hybrid retrieval according to the evolving context, achieves the best average performance across benchmarks. Notably, although PILOTRAG is not the strongest on the in-domain HotpotQA dataset in isolation, it consistently delivers robust performance across all datasets, leading to the highest overall accuracy. This indicates that PILOTRAG generalizes more effectively across diverse unseen datasets, balancing the complementary strengths of different retrieval modes.

These findings demonstrate that PILOTRAG not only integrates the benefits of text and graph retrieval but also learns to adaptively modulate its retrieval strategy, thereby achieving both higher robustness and stronger generalization compared to fixed retrieval modes.

## D EXPERIMENTS ON 7B LLMS

To further examine the scalability of our framework, we train a 7B version of PILOTRAG based on Qwen2.5-7B-Instruct (Yang et al., 2025b). Unlike the 3B experiments, which rely on the Contriever for dense retrieval, the 7B LLMs use the stronger NV-Embed-v2 (Lee et al., 2025a) embedding model. This difference allows us to assess how the retrieval policy behaves under a higher-quality text retriever and a more capable backbone LLM.

Table 6 presents detailed comparisons across both simple QA and multi-hop QA benchmarks. PILOTRAG shows substantial gains over all graph-based and multi-turn baselines on multi-hop reasoning tasks, outperforming strong methods such as Search-R1 (Jin et al., 2025) and R1-Searcher (Song et al., 2025) by a large margin. Notably, PILOTRAG achieves the best average EM and F1 scores among all 7B methods. Search-R1 remains strong on simple QA, which aligns

with its training distribution bias toward single-hop reasoning. In contrast, R1-Searcher, trained only on multi-hop data, performs competitively on complex tasks but falls behind on simple queries. PILOTRAG exhibits a more balanced behavior, while slightly behind Search-R1 on simple QA, it significantly surpasses both baselines on all multi-hop benchmarks. This suggests that RL with hybrid graph-text retrieval allows the model to generalize both within and beyond the multi-hop domain.

These results demonstrate that PILOTRAG scales effectively to larger LLMs and benefits from stronger retrieval components, achieving state-of-the-art performance on multi-hop QA tasks while maintaining competitive results on simple QA.

## E  CASE STUDY

To further illustrate how our training framework improves model behavior, we present several qualitative case studies comparing the outputs of the base model (Qwen2.5-3B-Instruct) before training and our proposed PILOTRAG after training, as shown in Tables 7–10. These cases reveal several representative issues of the base model and how our training addresses them:

**Irrelevant or spurious reasoning chains**. The base model often generated lengthy reasoning traces that wandered away from the actual question and were not grounded in retrieved evidence (Cases 1 & 3). In contrast, our trained model learns to anchor each reasoning step to retrieved content, reducing unnecessary speculation and improving logical coherence.

**Failure to understand or decompose complex questions**. The base model frequently struggled with multi-hop or compositional questions, failing to recognize their underlying structure or to plan effective retrieval steps (Cases 2 & 4). Through training, our model acquires the ability to analyze the question, break it down into subproblems, and issue precise retrieval queries for each part, which leads to more accurate and complete answers.

**Over-reliance on internal knowledge without validation**. The base model tended to rely on its parametric knowledge and give confident but unsupported answers, skipping the step of verifying correctness against external evidence (Case 3). Our model, by contrast, learns to cross-check candidate answers with retrieved documents before finalizing its response, which reduces hallucinations and increases factual accuracy.

Overall, after training, PILOTRAG shows clear improvements in its ability to understand complex questions, decompose them into manageable subproblems, and construct concise yet targeted retrieval queries that lead to well-grounded, correct answers.

Table 7: **Case study 1**. Before training, the model retrieved relevant evidence about both individuals but produced a logically inconsistent conclusion, incorrectly claiming they are not both singers. After training, it correctly integrates the retrieved evidence and arrives at the correct answer.

---

**Question**: Are Steve Perry and Leslie West both singers?

---

**Ground Truth**: Yes

---

**Before Training (Qwen2.5-3B-Instruct)**
`<think>` To determine if Steve Perry and Leslie West are both singers, I need to find information about their careers and musical roles. `</think>`
`<search> [passage]` Steve Perry Leslie West singers `</search>`
`<information>` Doc 1(Title: Leslie West Live!) Leslie West Live! is a live album by Leslie West, released in 1993.
Doc 2(Title: The Leslie West Band) The Leslie West Band is the third album released by American rock guitarist Leslie West... `</information>`

------------------------------------------------------------------------------------------

`<think>` From the search results, we can see that Leslie West is indeed a singer and guitarist. However, there is no direct mention of Steve Perry's musical career. Let's look for more information about Steve Perry. `</think>`
`<search> [passage]` Steve Perry singers `</search>`
`<information>` Doc 1(Title: Steve Perry (Oregon musician)) Stephen Henry Perry (born October 8, 1963) is an American musician, best known as lead singer, songwriter and rhythm guitarist for the Oregon ska-swing band the Cherry Poppin' Daddies... `</information>`

------------------------------------------------------------------------------------------

`<think>` From the search results, we can see that Steve Perry is a singer, but he is not known for being a member of a band. He is the lead singer of the band the Cherry Poppin' Daddies, which is a ska-swing band. Leslie West, on the other hand, is a founding member of the hard rock band Mountain and a guitarist and vocalist. `</think>`
`<answer>` No, Steve Perry and Leslie West are not both singers. Steve Perry is a singer, but he is not known for being a member of a band, while Leslie West is a singer and guitarist. `</answer>`

---

**After Training (PILOTRAG)**
`<think>` To determine if Steve Perry and Leslie West are both singers, I need to first identify who they are and then verify their profession. Let's start with identifying who they are. `</think>`
`<search> [graph]` Steve Perry `</search>`
`<information>` Doc 1(Title: Arrival (Journey album)) Arrival is the eleventh studio album by the American rock band Journey, released in the United States in 2001...The album was the band's first full-length studio album with new lead vocalist Steve Augeri, who replaced popular frontman Steve Perry... `</information>`

------------------------------------------------------------------------------------------

`<think>` From Doc 1, it's clear that Steve Perry is a lead vocalist for Journey. Let's now identify Leslie West. `</think>`
`<search> [graph]` Leslie West `</search>`
`<information>` Doc 1(Title: The Leslie West Band) The Leslie West Band is the third album released by American rock guitarist Leslie West... `</information>`

------------------------------------------------------------------------------------------

`<think>` From Doc 3, it's evident that Leslie West is a rock guitarist and vocalist. Now that we have identified both individuals, we can conclude whether they are singers. Since Steve Perry is a lead vocalist for Journey and Leslie West is a vocalist and guitarist, both are singers. `</think>`
`<answer>` Yes `</answer>`

Table 8: **Case study 2**. Before training, the model misunderstood the question and produced a long query, leading it to return only the band name without addressing the actual question. After training, it correctly interprets the question, decomposes it into sub-steps, and formulates concise, targeted queries that retrieve the necessary evidence to answer it.

---

**Question**: The View from the Bottom is the fifth studio album by an American rock band best known for what hit song?

---

**Ground Truth**: My Own Worst Enemy

---

**Before Training (Qwen2.5-3B-Instruct)**
`<think>` To answer this question, I need to identify the American rock band best known for a hit song and then find their fifth studio album. The album "The View from the Bottom" is the fifth studio album by this band. `</think>`
`<search> [graph]` The View from the Bottom American rock band hit song `</search>`
`<information>` Doc 1(Title: The View from the Bottom) The View from the Bottom is the fifth studio album by the American rock band Lit, released on June 19, 2012 through Megaforce Records... `</information>`
--------------------------------------------------------------------------------
`<answer>` Lit `</answer>`

---

**After Training (PILOTRAG)**
`<think>` To answer this question, I need to identify the American rock band whose fifth studio album is "The View from the Bottom" and then determine their best-known hit song. Let's start by identifying the band. `</think>`
`<search> [graph]` The View from the Bottom `</search>`
`<information>` Doc 1(Title: The View from the Bottom) The View from the Bottom is the fifth studio album by the American rock band Lit, released on June 19, 2012 through Megaforce Records... `</information>`
--------------------------------------------------------------------------------
`<think>` From Doc 1, it is clear that "The View from the Bottom" is by the band Lit. Now, I need to identify their best-known hit song. `</think>`
`<search> [graph]` Lit best known hit song `</search>`
`<information>` Doc 1(Title: Lit (band)) Lit is an American rock band, formed in 1995 in Fullerton, California. They are best known for their hit song "My Own Worst Enemy"... `</information>`
--------------------------------------------------------------------------------
`<think>` From Doc 1, it is evident that Lit's best-known hit song is "My Own Worst Enemy". Therefore, the answer to the question is: `</think>`
`<answer>` My Own Worst Enemy `</answer>`

Table 9: **Case study 3**. Before training, the model hallucinates incorrect internal knowledge, thus wrongly claiming that Johnny Pemberton played Bo Thompson in "That '70s Show" and that it was created by Steven Molaro. After training, it avoids hallucinations, correctly decomposes the question into sub-steps, and retrieves the right evidence to identify the creator as Justin Spitzer.

---

**Question**: Who created the NBC sitcom that Johnny Pemberton appears in as the character Bo Thompson?

---

**Ground Truth**: Justin Spitzer

---

**Before Training (Qwen2.5-3B-Instruct)**
`<think>` To answer this question, I need to identify the NBC sitcom that Johnny Pemberton appeared in as the character Bo Thompson. Let's start with the reasoning: 1. Johnny Pemberton is known for his role as Bo Thompson in the NBC sitcom "That '70s Show." 2. "That '70s Show" aired from 1998 to 2006. 3. The show was created by Steven Molaro, who wrote the pilot and served as showrunner. Now, I will search for more detailed information about the creator of "That '70s Show" to confirm the answer. `</think>`
`<search>` `[passage]` Steven Molaro That '70s Show creator `</search>`
`<information>` Doc 1(Title: Bigfoot (TV series)) Bigfoot is an American animated series television pilot created by Seth Rogen, Matt McKenna, and Evan Goldberg... `</information>`

--------------------------------------------------------------------------------

`<think>` Based on the search results, the information about Steven Molaro and "That '70s Show" is not found in the documents. However, the documents do not contain any information about Johnny Pemberton or the show "That '70s Show." Since the documents do not provide the necessary information, I will use graph-based retrieval to find the creator of "That '70s Show." `</think>`
`<search>` `[graph]` That '70s Show creator `</search>`
`<information>` Doc 1(Title: Parent show) Refers to a TV series that has spawned a spin-off... `</information>`

--------------------------------------------------------------------------------

`<think>` The search results do not provide the creator of "That '70s Show." Given the lack of relevant information, I will conclude based on the initial reasoning: `</think>`
`<answer>` Steven Molaro `</answer>`

---

**After Training (PILOTRAG)**
`<think>` To answer this question, I need to first identify the NBC sitcom Johnny Pemberton appears in as the character Bo Thompson. Then, I'll find out who created that show. Let's start with the sitcom. `</think>`
`<search>` `[graph]` Johnny Pemberton Bo Thompson `</search>`
`<information>` Doc 1(Title: Johnny Pemberton) Johnny Pemberton (born in 1981) is an American actor and comedian from Rochester, Minnesota. He is best known for his role as the titular Šon of Zornïn the short-lived Fox sitcom, and has also appeared as the recurring character Bo Thompson in the NBC sitcom "Superstore"... `</information>`

--------------------------------------------------------------------------------

`<think>` From Doc 1, it's clear that Johnny Pemberton appeared as Bo Thompson in "Superstore". Now, I need to find out who created "Superstore". `</think>`
`<search>` `[graph]` Superstore creator `</search>`
`<information>` Doc 1(Title: Superstore (TV series)) Superstore is an American single-camera sitcom television series that premiered on NBC on November 30, 2015. The series was created by Justin Spitzer... `</information>`

--------------------------------------------------------------------------------

`<think>` From Doc 1, it's evident that "Superstore" was created by Justin Spitzer. Therefore, the answer to the question is: `</think>`
`<answer>` Justin Spitzer `</answer>`

Table 10: **Case study 4**. Before training, the model failed to answer the question due to a lack of reasoning and question decomposition. After training, the model correctly decomposed the question into two steps, i.e., finding the surfer with the given birth date and then verifying his victory.

---

**Question**: What American professional Hawaiian surfer born 18 October 1992 won the Rip Curl Pro Portugal?

---

**Ground Truth**: John John Florence

---

**Before Training (Qwen2.5-3B-Instruct)**
`<think>` To answer this question, I need to find information about a professional Hawaiian surfer born on 18 October 1992 who won the Rip Curl Pro Portugal. Let's start with a search using both dense and graph-based retrieval methods. `</think>`
`<search> [graph][passage]` American professional Hawaiian surfer born 18 October 1992 won the Rip Curl Pro Portugal `</search>`
`<information>` Doc 1(Title: MEO Rip Curl Pro Portugal) MEO Rip Curl Pro Portugal...The defending champion is the 2016 WSL champion, John John Florence who won the event...
Doc 3(Title: Coco Ho) Coco Malia Camille Hapaikekoa Ho (born April 28, 1991) is a professional Hawaiian surfer born in Honolulu, Hawaii. `</information>`
------------------------------------------------------------------------------------------
`<think>` From the search results, it seems there is no information about a surfer born on 18 October 1992 winning the Rip Curl Pro Portugal. However, Coco Ho is mentioned in Doc 3, and she was born on April 28, 1991, not October 18, 1992. Given the discrepancy, let's refine our search to focus on the Rip Curl Pro Portugal winner. `</think>`
`<search> [graph][passage]` Rip Curl Pro Portugal winner born 18 October 1992 `</search>`
`<information>` Doc 1(Title: MEO Rip Curl Pro Portugal)...`</information>`
------------------------------------------------------------------------------------------
`<think>` The search results still do not provide the specific information we need. Let's try a more direct search focusing on the Rip Curl Pro Portugal winner. `</think>`
`<search> [graph][passage]` Rip Curl Pro Portugal winner 18 October 1992 `</search>`
`<information>` Doc 1(Title: MEO Rip Curl Pro Portugal)...`</information>`
------------------------------------------------------------------------------------------
`<think>` After reviewing the documents, it appears there is no surfer born on 18 October 1992 who has won the Rip Curl Pro Portugal. The defending champion in 2016 was John John Florence, but he was born on 14 March 1992, not 18 October 1992. `</think>`
`<answer>` No surfer born on 18 October 1992 has won the Rip Curl Pro Portugal. `</answer>`

---

**After Training (PILOTRAG)**
`<think>` To answer this question, I need to identify the American professional Hawaiian surfer born on 18 October 1992 who won the Rip Curl Pro Portugal. Let's start by finding the surfer born on that date. `</think>`
`<search> [graph]` born 18 October 1992 American professional Hawaiian surfer `</search>`
`<information>` Doc 1(Title: Coco Ho) Coco Malia Camille Hapaikekoa Ho...
Doc 2(Title: John John Florence) John John Alexander Florence (born 18 October 1992) is an American professional surfer... `</information>`
------------------------------------------------------------------------------------------
`<think>` From the retrieved documents, Doc 2 clearly mentions John John Florence, who was born on 18 October 1992 and is an American professional surfer. Now, I need to confirm if he won the Rip Curl Pro Portugal. `</think>`
`<search> [graph]` John John Florence Rip Curl Pro Portugal `</search>`
`<information>` Doc 1(Title: Supertubos) The Supertubos (Portuguese for S̈upertubes)̈ is a beach located in Peniche, Portugal...
Doc 2(Title: MEO Rip Curl Pro Portugal)... `</information>`
------------------------------------------------------------------------------------------
`<think>` From the retrieved documents, Doc 2 clearly mentions John John Florence as the defending champion and winner of the MEO Rip Curl Pro Portugal in 2016. Therefore, the answer to the question is: `</think>`
`<answer>` John John Florence `</answer>`

