# OpenReview forum: "PilotRAG: Teaching LLMs Multi-Turn Hybrid RAG via Reinforcement Learning"
_ICLR.cc/2026/Conference — ICLR 2026 Conference Withdrawn Submission_

### Official Review · Reviewer_4PR3 · 2025-10-20

**Soundness:** 3
**Presentation:** 2
**Contribution:** 2
**Rating:** 2
**Confidence:** 4

**Summary:**

This paper proposes a RL method that enables LLMs to interleave reasoning and retrieval actions across both text and graph sources. The model uses a two-stage GRPO-based training procedure to balance answer accuracy and retrieval efficiency. Experiments on multiple QA benchmarks claim significant gains over previous RAG and graph-RAG systems.

**Strengths:**

1. The paper attempts to unify text and graph retrieval with reinforcement learning, producing an end-to-end framework that is well-articulated.

2. The model shows quantitative improvements over existing multi-turn RAG baselines such as Search-R1 and HippoRAG v2 across multiple benchmarks.

**Weaknesses:**

1. The main technical idea is a straightforward combination of existing retrieval types (text + graph) within a standard GRPO framework. There is no genuinely new retrieval or RL formulation; the method feels incremental and engineering-oriented.

2. The so-called “theoretical analysis” in the appendix is a bit trivial as it merely re-states GRPO variance reduction logic with algebraic reformatting, adding no formal insight or new guarantees.

3. Although efficiency is emphasized as a reward signal, the paper provides no real complexity or runtime comparison with baselines beyond qualitative plots. There is no measurement of wall-clock savings, GPU utilization, or scalability under large graphs.

4. The ablation only removes the efficiency reward; no experiment isolates hybrid retrieval, GRPO itself, or the contribution of multi-turn structure.

5. Some other baselines, such as R1-Searcher and SimpleDeepSearcher, are not compared to.

**Questions:**

1. What new insight does “hybrid RRF fusion” bring over existing ensemble retrieval methods like reciprocal rank fusion?

2. Can the authors demonstrate any generalization beyond the closed HotpotQA-like domains, or is this yet another narrow QA-specific RL setup?

---

> ### Author Response · Authors · 2025-11-19
> **Response to Reviewer 4PR3 (Part 1)**
>
> Thank you for the careful and constructive review. We appreciate the opportunity to clarify our contributions and to improve the manuscript. Below we respond point-by-point.
>
> > **Weakness 1**. The main technical idea is a straightforward combination of existing retrieval types (text + graph) within a standard GRPO framework.
>
> Thank you for raising this point. Our goal is not to introduce a new retrieval primitive or a new RL formulation, but rather to  **integrate hybrid retrieval into an iterative reasoning process and make it trainable under efficiency-aware RL signals**:
>
> * **Hybrid retrieval in an RL-trained routing loop.** While hybrid combinations of text- and graph-based retrieval have been considered in prior works, PilotRAG integrates these heterogeneous retrieval modalities into a **single** **learned policy** that is trained end-to-end with an objective that balances  **accuracy and retrieval cost**.
> * **Efficiency-aware RL (Stage 2) targeted at Graph-based RAG.** We design the efficiency reward specifically to address the practical bottleneck of multi-turn graph retrieval (graph construction and repeated graph queries). This is not merely a generic latency penalty but is incorporated in a way that encourages the policy to avoid unnecessary graph accesses while preserving answer quality.
> * **Empirical effect in constrained-resource settings.** We show that enabling efficiency-aware training yields meaningful inference-time reductions while improving performance in our benchmarks.
>
> We therefore view PilotRAG as a contribution that advances **practical, deployable** RAG systems by making graph-based retrieval both effective and cost-aware, rather than proposing a wholly new mathematical formalism.
>
> > **Weakness 2**. The so-called "theoretical analysis" in the appendix is a bit trivial as it merely re-states GRPO variance reduction logic with algebraic reformatting, adding no formal insight or new guarantees.
>
> Appendix A focuses on explaining  **why the proposed efficiency reward interacts well with GRPO**, and why certain averaging strategies are appropriate in this context. This necessarily involves restating core GRPO equations to show how our reward integrates into its variance-reduced structure. The key insight is that  **batch-wise average retrieval time provides a stable and effective learning signal under GRPO**, whereas other RL algorithms, e.g., PPO, may over-penalize queries that naturally require longer retrieval chains. The purpose is therefore clarifying why our reward design is compatible with GRPO.
>
> > **Weakness 3**. Although efficiency is emphasized as a reward signal, the paper provides no real complexity or runtime comparison with baselines beyond qualitative plots.
>
> The retrieval-time bar charts in Figure 2 are in fact  **quantitative**: every bar directly reflects the wall-clock retrieval time of the model and thus effectively captures the actual runtime savings. Similarly, reduced retrieval time leads to proportional reductions in GPU usage during both retrieval and training.
>
> > **Weakness 4**. The ablation only removes the efficiency reward; no experiment isolates hybrid retrieval, GRPO itself, or the contribution of multi-turn structure.
>
> We provide several forms of ablation across the paper and appendix:
>
> * **Retrieval-mode ablation**: Appendix C reports performance under different retrieval modes, including hybrid retrieval.
> * **GRPO contribution**: Table 3 presents model results without GRPO training.
> * **Multi-turn structure**: The comparison among HippoRAG (single-turn), PilotRAG w/o training illustrates the role of iterative reasoning in some extent.
>
> In short,
>
> * Graph retrieval provides the majority of gains among retrieval modes.
> * Multi-turn reasoning without RL offers modest improvements (~2%) because the 3B model cannot reliably complete long reasoning chains.
> * Incorporating multi-turn reasoning with GRPO provides a large performance boost (~20%), reflecting the synergy between iterative structure and RL training.

---

> ### Author Response · Authors · 2025-11-19
> **Response to Reviewer 4PR3 (Part 2)**
>
> > **Weakness 5**. Some other baselines, such as R1-Searcher and SimpleDeepSearcher, are not compared to.
>
> R1-Searcher is indeed an important baseline and is discussed in our related work. SimpleDeepSearcher targets a different problem setting, i.e., web search over external engines rather than knowledge-base retrieval, making it less aligned with our setting.
>
> For R1-Searcher, their released model was only available at the 7B&8B scale, whereas our main experiments centered on 3B models, making a direct comparison initially inappropriate. We have conducted additional experiments on a 7B version of PilotRAG using the same Qwen2.5 backbone as R1-Searcher, and details are available in the **Appendix D** (Experiments on 7B LLMs). Below are the results in comparison to R1-Searcher (part of Table 6 of the revised paper):
>
> | Method             | PopQA EM       | PopQA F1       | NQ EM          | NQ F1          | Hotpot EM      | Hotpot F1      | 2Wiki EM       | 2Wiki F1       | MuSiQue EM     | MuSiQue F1     | Avg EM         | Avg F1         |
> | ------------------ | -------------- | -------------- | -------------- | -------------- | -------------- | -------------- | -------------- | -------------- | -------------- | -------------- | -------------- | -------------- |
> | R1-Searcher        | 28.4           | 41.0           | 41.6           | 52.2           | 46.6           | 56.7           | 41.7           | 49.0           | 29.3           | 37.6           | 37.5           | 47.3           |
> | **PilotRAG** | **50.6** | **56.4** | **51.5** | **60.4** | **60.8** | **72.5** | **57.1** | **64.6** | **39.6** | **49.3** | **51.9** | **60.6** |
>
> > **Question 1**. What new insight does "hybrid RRF fusion" bring over existing ensemble retrieval methods like reciprocal rank fusion?
>
> The hybrid RRF fusion in our method is not intended to introduce a fundamentally new ensemble technique, but to provide a **lightweight unification** that allows text and graph retrieval results to be merged in a principled manner. The contribution lies in enabling the model to consult from both retrieval channels within a single iterative decision loop, rather than proposing a new ensemble formula.
>
> > **Question 2**. Can the authors demonstrate any generalization beyond the closed HotpotQA-like domains, or is this yet another narrow QA-specific RL setup?
>
> Our design is indeed motivated by the RAG scenario, where the model must reason over both textual and relational knowledge sources. This naturally places our evaluation within the standard knowledge-intensive QA benchmarks.
>
> However, the  **RL framework itself is not restricted to HotpotQA-style tasks**. The formulation, i.e., multi-turn reasoning, learned retrieval decisions, and efficiency-aware GRPO optimization, does not assume any specific domain or dataset structure. As long as the environment provides (i) retrieval actions, (ii) a verifiable task reward, and (iii) a measurable efficiency signal, the same framework can, in principle, be applied to other settings or even non-QA tasks that involve iterative tool invocation or search.

---

### Official Review · Reviewer_vAcV · 2025-10-29

**Soundness:** 1
**Presentation:** 3
**Contribution:** 2
**Rating:** 2
**Confidence:** 4

**Summary:**

This paper mainly extends graph-text hybrid RAG into multi-turn scenarios. It utilizes RL training by constructing performance rewards and efficiency rewards. It conducts experiments on several datasets.

**Strengths:**

The methods are clearly stated. Experiments validate the methods on multiple datasets.

**Weaknesses:**

1. I think the innovation of the paper's methods is limited. In Section 3.1, the overall model architecture is very similar to previous iterative RAGs, such as Search-R1, except that the retrieval methods are expanded to hybrid ones. However, these retrieval methods are also implemented in previous work, as shown in Section 3.1.2. Regarding RL optimization, Stage 1 simply measures the reward for task completion. Although Stage 2 introduces total search time as a loss, this does not align with the core motivation in the abstract, and does not provide targeted optimization for Graph-RAG. Therefore, I consider this paper's approach to be incremental.
2. I think Algorithm 1 needs to be optimized because it's unclear. While I can generally understand what it describes after reading the subsequent sections, some of the symbols used in this algorithm are not explained, making it a bit confusing.
3. Experiments were conducted only on the Qwen 2.5-3B model. Validation on 7B and higher models is strongly recommended.
4. In the provided anonymous Github repository, under "Step 1: Start Services," it is mentioned that Llama-3.1-8B-Instruct was used in the experiments, but this is not mentioned in the paper. Please provide more explanations.

**Questions:**

See weaknesses above. Please respond to these cons.

---

> ### Author Response · Authors · 2025-11-19
> **Response to Reviewer vAcV (Part 1)**
>
> Thank you very much for the detailed and constructive feedback. We sincerely appreciate the opportunity to clarify the design motivations and improve the presentation of our work.
>
> > **Weakness 1.** The innovation of the paper's methods is limited.
>
> We appreciate your thoughtful assessment. Below we clarify the design choices and motivations behind our method:
>
> * **On similarity to prior iterative RAGs.**
>
>   We agree that our overall workflow follows the established *think–retrieve–revise* paradigm commonly used in iterative RAG systems. This paradigm has been widely adopted in prior works [1,2,3], and forms the basis of most multi-step retrieval-and-reasoning pipelines. Our contribution lies in extending this paradigm to  **hybrid retrieval with graph reasoning**, and enabling it to be  **optimized through RL under both accuracy and efficiency objectives**.
> * **On RL Stage 1 (Outcome-oriented reward).**
>
>   Stage 1 aims to provide the model with  **foundational reasoning ability**, especially because we adopt a relatively small backbone. Outcome-only rewards have been shown to be effective for this purpose in Search-R1 and R1-Searcher [3]. Our design follows this established practice rather than introducing unnecessary complexity at the early stage.
> * **On RL Stage 2 (Accuracy–efficiency reward).**
>
>   The motivation for Stage 2 is directly tied to a core challenge in GraphRAG:  graph construction and multi-hop graph retrieval are computationally expensive , particularly in iterative settings. Incorporating the retrieval-time term into the reward encourages the model to  reason more efficiently, which is precisely a targeted optimization for graph-based RAG.
>
> [1] Interleaving retrieval with chain-of-thought reasoning for knowledge-intensive multi-step questions. ACL 2023.
>
> [2] Search-o1: Agentic search-enhanced large reasoning models. EMNLP 2025.
>
> [3] R1-Searcher: Incentivizing the Search Capability in LLMs via Reinforcement Learning. arXiv preprint arXiv:2503.05592 (2025).
>
> > **Weakness 2**. Algorithm 1 needs to be optimized because it's unclear.
>
> Thank you for pointing out the clarity issue. We have **revised Algorithm 1** **and the accompanying explanation in Section 3.1.1** to ensure all symbols are clearly defined and that the workflow is easier to follow.
>
> > **Weakness 3**. Experiments were conducted only on the Qwen 2.5-3B model. Validation on 7B and higher models is strongly recommended.
>
> Thank you for this valuable suggestion. We have conducted additional experiments using  **Qwen2.5-7B-Instruct**, and compared PilotRAG-7B to the 7B versions of Search-R1, R1-Searcher, Search-o1, and HippoRAG v2. The results are now included in the **Appendix D** of the revised paper. Below are the main experimental results ($^*$ represents in-domain datasets):
>
> | Method             | PopQA EM       | PopQA F1       | NQ EM           | NQ F1           | Hotpot EM       | Hotpot F1       | 2Wiki EM       | 2Wiki F1       | MuSiQue EM     | MuSiQue F1     | Avg EM         | Avg F1         |
> | :----------------- | -------------- | -------------- | --------------- | --------------- | --------------- | --------------- | -------------- | -------------- | -------------- | -------------- | -------------- | -------------- |
> | HippoRAG v2        | 27.0           | 37.9           | 8.1             | 27.2            | 27.4            | 46.1            | 16.8           | 34.7           | 12.3           | 24.0           | 18.3           | 34.0           |
> | Search-o1          | 4.7            | 7.2            | 18.1            | 27.5            | 13.5            | 19.1            | 6.4            | 7.9            | 2.9            | 7.7            | 9.1            | 13.9           |
> | R1-Searcher        | 28.4           | 41.0           | 41.6            | 52.2            | 46.6*           | 56.7*           | 41.7*          | 49.0*          | 29.3           | 37.6           | 37.5           | 47.3           |
> | Search-R1          | **51.3** | **57.1** | **56.8*** | **65.3*** | 51.0*           | 62.0*           | 51.8           | 58.9           | 32.0           | 40.8           | 48.6           | 56.8           |
> | **PilotRAG** | 50.6           | 56.4           | 51.5            | 60.4            | **60.8*** | **72.5*** | **57.1** | **64.6** | **39.6** | **49.3** | **51.9** | **60.6** |

---

> ### Author Response · Authors · 2025-11-19
> **Response to Reviewer vAcV (Part 2)**
>
> > **Weakness 4**. Llama-3.1-8B-Instruct was used in the experiments, but this is not mentioned in the paper.
>
> We appreciate the opportunity to clarify this point. The Llama-3.1-8B-Instruct model is  **only used for Information Extraction (IE) during graph construction**, not during the reasoning stage. Specifically, it extracts entities and triplets from corpus passages to build the knowledge graph that HippoRAG v2 (and thus PilotRAG) consults.
>
> This follows the implementation of HippoRAG v2, which uses much larger models (i.e., Llama-3.3-70B-Instruct / GPT-4o-mini) for IE. For our 3B PilotRAG experiments, we opted for the lighter Llama-3.1-8B-Instruct to reduce computational cost and to improve reproducibility. As for the additional experiments conducted on Qwen2.5-7B, we adopt the same Llama-3.3-70B-Instruct IE model that HippoRAG v2 uses to provide a graph with higher quality.

---

> ### Comment · Reviewer_vAcV · 2025-11-24
>
> Thanks for your rebuttal. I have raised my rating to 4, but I think it is still on the borderline of rejection.

---

> > ### Author Response · Authors · 2025-11-25
> >
> > Thank you for raising your score. We sincerely appreciate your endorsement!

---

### Official Review · Reviewer_Gxyx · 2025-11-01

**Soundness:** 3
**Presentation:** 3
**Contribution:** 3
**Rating:** 4
**Confidence:** 3

**Summary:**

This paper proposes to improve current RL-based multi-turn RAG method with hybrid search. Specifically, each search action can be selected from passage retrieval, graph-based retrieval, or a hybrid of both. Then, the model is trained with GRPO with a 2-stage method. In the first stage, the model is trained only to achieve higher accuracy, and in the second stage, the model is also encouraged to be more efficient measured by time.

**Strengths:**

- Combining text-based and graph-based retrieval in the RL-based multi-turn RAG in principle should help the model achieve higher performance with more adaptive retrieval results
- Experiment results show the method achieves strong performance, especially the stage-2 training shows it can improve efficiency while not sacrificing performance
- Paper writing is clear

**Weaknesses:**

- The proposed method seems to achieve promising performance on multi-hop QA datasets but the performance on simple QA is clearly hurt with hybrid search (e.g., comparing search-r1 with PILOTRAG w/o efficiency training). Therefore, there should be more analysis on why hybrid strategy fails on simple QA and how to mitigate it.
- Some efficiency comparison could be beneficial to show the effects of stage 2 training other than just improving performance.

**Questions:**

- When testing the model, what is the ratio of the 3 search actions on different datasets?

---

> ### Author Response · Authors · 2025-11-19
> **Response to Reviewer Gxyx**
>
> Thank you very much for the thoughtful and constructive feedback. We sincerely appreciate the opportunity to clarify these points and further improve the presentation of our work.
>
> > **Weakness 1.** The proposed method seems to achieve promising performance on multi-hop QA datasets but the performance on simple QA is clearly hurt with hybrid search.
>
> We appreciate this insightful observation. The difference in performance on simple QA can be largely attributed to the **training data distribution** rather than an inherent limitation of the hybrid strategy itself. PilotRAG is trained on  **10k samples from HotpotQA**, which predominantly contain multi-hop questions. In contrast, Search-R1 is trained on **170k samples spanning both simple and multi-hop datasets** (NQ + HotpotQA), giving it substantially more exposure to single-hop patterns. Our additional 7B experiments further support this explanation. R1-Searcher, which is also trained only on multi-hop data and with a data size similar to that of PilotRAG, exhibits poor performance on simple QA tasks. In comparison, PilotRAG remains competitive under the same training data conditions.
>
> | Method                  | PopQA EM       | PopQA F1       | NQ EM          | NQ F1          | Hotpot EM      | Hotpot F1      | 2Wiki EM       | 2Wiki F1       | MuSiQue EM     | MuSiQue F1     | Avg EM         | Avg F1         |
> | ----------------------- | -------------- | -------------- | -------------- | -------------- | -------------- | -------------- | -------------- | -------------- | -------------- | -------------- | -------------- | -------------- |
> | R1-Searcher (7B)        | 28.4           | 41.0           | 41.6           | 52.2           | 46.6           | 56.7           | 41.7           | 49.0           | 29.3           | 37.6           | 37.5           | 47.3           |
> | **PilotRAG** (7B) | **50.6** | **56.4** | **51.5** | **60.4** | **60.8** | **72.5** | **57.1** | **64.6** | **39.6** | **49.3** | **51.9** | **60.6** |
>
> We adopted a smaller training set mainly due to  **computational considerations**: both graph construction and graph retrieval incur significant overhead during RL training. Despite this constraint, PilotRAG with efficiency reward still performs competitively on simple QA benchmarks.
>
> > **Weakness 2**. Some efficiency comparison could be beneficial to show the effects of stage 2 training other than just improving performance.
>
> Thank you for pointing this out. We provide a comparison of retrieval latency between PilotRAG and its variant without efficiency reward in Figure 2. As elaborated in Section 4.3.2, incorporating the efficiency reward leads to:
>
> 1. **Reduced retrieval time** across all datasets, and
> 2. **Improved overall QA performance**, indicating that efficiency-aware training not only accelerates inference but also guides the model toward more targeted and effective retrieval strategies.
>
> We will highlight these findings more explicitly to strengthen the connection between Stage 2 training and its efficiency benefits.
>
> > **Question 1**. When testing the model, what is the ratio of the 3 search actions on different datasets?
>
> Thank you for the question. During testing, we find that the retrieval policy does not converge to a fixed ratio but adapts to the underlying model–retriever configuration. For the  **3B model with Contriever**, the policy leans toward graph retrieval because Contriever has weaker recall on entity-centric queries, making graph retrieval a more reliable source of multi-hop evidence during RL training. In contrast, the **7B model with NV-Embed-V2** exhibits a higher use of hybrid retrieval, as the stronger embedding model improves text-level recall and allows hybrid fusion to yield consistently higher rewards.
>
> These patterns do not contradict the motivation of PilotRAG. The goal is not to enforce a specific retrieval distribution but to let the policy learn  **task-adaptive routing**. The different ratios simply reflect the most effective retrieval balance discovered by the policy under each configuration.

---

> ### Comment · Reviewer_Gxyx · 2025-11-26
>
> Thanks for the response and additional experiments. I understand that different training data may cause the issue claimed here.
>
> However, I insist on my question about the ratio of different knowledge sources being routed to. I didn't expect there should be a "specific fixed ratio", sorry if the authors are confused. I just expect to see some analysis for people to understand how the routing decision affects the retrieval behavior. Thanks to other reviewers who pointed out that all the retrievals in case studies are graph retrieval. I believe some analysis here could be essential to better understand the proposed method.

---

### Official Review · Reviewer_6eey · 2025-11-04

**Soundness:** 2
**Presentation:** 3
**Contribution:** 2
**Rating:** 2
**Confidence:** 4

**Summary:**

This paper tackles the limitation of one-shot retrieval in both traditional and graph RAG systems by fine-tuning a multi-step hybrid RAG policy LLM - PilotRAG. It features a two-stage RLVR pipeline by first performing outcome-oriented training then accuracy-efficiency training . By training a policy LLM, the model learns to interleave the three different retrieval actions and reasoning. After optimizing on hotpotQA, PilotRAG outperforms recent graph RAG and multi-RAG on same generator model (Qwen2.5) across multiple different datasets.

**Strengths:**

1. This paper proposes an effective and practical reward system to help policy model balance between the reasoning efficiency and tool-use in a hybrid RAG setting (both textual and relational information are presented).

**Weaknesses:**

1. Some of the technical details are not described in the paper, such as generalization of the graph construction algorithm (re-uses hipporag graph).

2. The paper is not aware of some important baselines that are published on the same topic: hyrbid text and graph rag. For example, HybGRAG[1] introduces hyrbid graph retrieval: choice between vector search and one-hop graph search with a pre-trained LLM (Sonnet3.5). These related work are not discussed, which also makes the claim in the abstract "graph-based RAG remains limited to one-shot retrieval" unappropriate.

3. In case study, all of the Pilot-RAG examples selects the graph retrieval, neither text nor hybrid retrieval are selected. It seems the performance improvements are from the graph retrieval instead of the RL.

[1] HybGRAG: Hybrid Retrieval-Augmented Generation on Textual and Relational Knowledge Bases, ACL 25'

**Questions:**

1. Does Pilot-RAG generalize to different graphs such as thoese in CRAG [1] and hyperlink graphs?

2. The training dynamics showing non-convergent #actions and #response length. Does this mean model is not converged or trained properly since no emerging reasoning is observed as other R1* work.

3. Please refer to my other questions in weaknesses.


[1] CRAG - Comprehensive RAG Benchmark

---

> ### Author Response · Authors · 2025-11-19
> **Response to Reviewer 6eey**
>
> Thank you very much for the thoughtful and constructive feedback! We treasure the opportunity to address your concerns and improve our work.
>
> > **Weakness 1 & Question 1**. The generalization of the graph construction algorithm.
>
> PilotRAG is designed as a  reasoning model , while the underlying graph-based RAG module is fully modularized and treated as a pluggable retriever. This decoupling allows PilotRAG to work with any graph retrieval backend, including those used in hyperlink graphs. Our current choice of HippoRAG stems from its strong empirical performance, but the framework itself does not rely on HippoRAG-specific assumptions.
>
> Indeed, the CRAG benchmark adopts a similar abstraction by exposing a mock KG API that isolates graph retrieval from reasoning. PilotRAG fits naturally into this modular paradigm and can generalize to different graph structures accordingly.
>
> > **Weakness 2**. The paper is not aware of some important baselines that are published on the same topic.
>
> Thank you for highlighting this important point. Following your suggestion, we have **updated the Related Work section** to include a discussion of HybGRAG and **revised the Abstract to avoid overstating prior limitations**. Specifically, we have modified "graph-based RAG remains limited to one-shot retrieval" to "existing graph-based and hybrid RAG methods generally rely on fixed or handcrafted multi-turn retrieval procedures rather than an RL-trained policy".
>
> While HybGRAG is highly relevant, it differs from our work in several key aspects:
>
> 1. **Efficiency considerations** are not addressed in HybGRAG, which is a central focus in PilotRAG.
> 2. Its graph retrieval component relies on  **triplet matching over a KG**, which is relatively limited compared to modern graph search algorithms.
> 3. HybGRAG does **not learn a reasoning/routing policy**, its retrieval mode selection is rule-based (based on whether an entity/relation is detected in the query), whereas PilotRAG trains a reasoning policy through RL.
>
> We also note that HybGRAG's source code has not been released, which unfortunately prevents us from including it as a reproducible baseline. Nonetheless, we ensure that its conceptual relation to our method is clearly discussed.
>
> > **Weakness 3**. It seems the performance improvements are from the graph retrieval instead of the RL.
>
> We agree that graph retrieval is an important component, however, the empirical results indicate that  **RL contributes substantial additional gains**. Compared with:
>
> * **HippoRAG v2 (graph-only baseline):** PilotRAG improves EM/F1 by  **over 20% on average**.
> * **Search-R1 (multi-turn dense retrieval baseline trained by RL):** PilotRAG further improves EM/F1 by  **6.6 / 6.4**, despite Search-R1 already using multi-turn retrieval.
>
> These improvements demonstrate that performance improvements are from both graph retrieval and RL, and even from RL more.
>
> > **Question 2.** Does model is not converged or trained properly since no emerging reasoning is observed as other R1* work.
>
> We apologize for any confusion caused by Figure 3. The model  **does exhibit emerging reasoning behavior throughout training** , as discussed in Section 4.3.3. The dynamics of action counts and response length reflect the model learning to construct more elaborate reasoning trajectories with training unfolds.
>
> We trained for 1 epoch based on early experiments in which extending to ~2 epochs did not yield further improvement and the curve shows that the model effectively  **converged within 1 epoch**. This is also evidenced in Figure 3, where rewards plateau toward the end. Thus, the observed patterns do not indicate failed convergence but rather the natural progression of RL training for this task.

---

### Note · Authors · 2025-11-30

I have read and agree with the venue's withdrawal policy on behalf of myself and my co-authors.